# Genomics of *Klebsiella pneumoniae* Species Complex Reveals the Circulation of High-Risk Multidrug-Resistant Pandemic Clones in Human, Animal, and Environmental Sources

**DOI:** 10.3390/microorganisms10112281

**Published:** 2022-11-17

**Authors:** Sergio Morgado, Erica Fonseca, Ana Carolina Vicente

**Affiliations:** Laboratory of Molecular Genetics of Microorganisms, Oswaldo Cruz Institute, Av. Brasil, 4365—Manguinhos, Rio de Janeiro 21040-900, Brazil

**Keywords:** epidemiology, Brazil, hypervirulence, resistome, environment, animal, siderophores, yersiniabactin, colibactin, T6SS

## Abstract

The *Klebsiella* species present a remarkable genetic and ecological diversity, being ubiquitous in nature. In particular, the *Klebsiella pneumoniae* species complex (KpSC) has emerged as a major public health threat in the world, being an interesting model to assess the risk posed by strains recovered from animals and the environment to humans. We therefore performed a genomic surveillance analysis of the KpSC using every public genome in Brazil, aiming to show their local and global relationships, and the connectivity of antibiotic resistance and virulence considering human, animal, and environmental sources. The 390 genomes from distinct sources encompassed the *K. pneumoniae*, *Klebsiella quasipneumoniae* subsp. *quasipneumoniae, Klebsiella quasipneumoniae* subsp. *similipneumoniae, Klebsiella variicola* subsp. *variicola*, *Klebsiella variicola* subsp. *tropica*, and *Klebsiella grimontii* species and subspecies. *K. pneumoniae* harbored dozens of antibiotic resistance genes, while most of the genomes belong to the high-risk pandemic CC258 occurring in humans, animals, and the environment. In *K. pneumoniae* ST11, a high prevalence of the virulence determinants yersiniabactin, colibactin, and T6SS was revealed in association with multi-drug resistance (MDR), including carbapenem resistance. A diversity of resistance genes is carried by plasmids, some shared between strains from different STs, regions, and sources. Therefore, here were revealed some factors driving the success of KpSC as a pathogen.

## 1. Introduction

The *Klebsiella* species (spp.) are ubiquitous, dispersed in natural environments such as soil, water, plants, and livestock, and may eventually be associated with both human and animal infections. This bacteria group presents a remarkable genetic and ecological heterogeneity, representing, therefore, a pertinent taxon for assessing the risk to public health posed by animal and environmental reservoirs [1]. Among the *Klebsiella* genus, the *Klebsiella pneumoniae* species is, so far, the best-studied species, mainly due to its frequent association with community-acquired and nosocomial outbreaks. Moreover, *K. pneumoniae* is included in the ESKAPE group due to its association with the spread and development of multi-drug resistance (MDR), being considered one of the major public health challenges worldwide [2].

All members of the *K. pneumoniae* species complex (KpSC), which includes *K. pneumoniae*, *Klebsiella quasipneumoniae* subsp. *quasipneumoniae*, *Klebsiella quasipneumoniae* subsp. *similipneumoniae*, *Klebsiella variicola* subsp. *variicola*, and *Klebsiella variicola* subsp. *tropica*, can be recovered from the environment, as well as from human and animal infections, showing resistance to multiple drugs and virulent phenotypes worldwide. [3,4,5,6,7,8,9]. These species, especially *K. pneumoniae*, can develop multidrug resistance to all antibiotic classes, including carbapenems, through gene acquisition via horizontal gene transfer [2]. β-lactams are antibiotics widely used around the world in the treatment of *Klebsiella* infections, such as penicillins, cephalosporins, and carbapenems. However, some strains develop resistance to these antibiotics by acquiring plasmid-mediated extended-spectrum β-lactamases (ESBL), AmpC, and carbapenemase genes, such as *bla*_CTX-M-15_, *bla*_SHV_, *bla*_TEM_, *bla*_DHA_, *bla*_CMY_, *bla*_KPC_, *bla*_OXA-48_, *bla*_VIM_, *bla*_IMP_, and *bla*_NDM_, being considered one of the greatest threats to global health [2]. The acquisition of mobile genetic elements has also led to the development of hypervirulent Kp (hvKp) that carry plasmids and integrative conjugative elements (ICEs) encoding several important virulence determinants, such as siderophores (salmochelin (*iro*), aerobactin (*iuc*), and yersiniabactin (*ybt*)), the colibactin toxin (*clb)* and/or genes responsible for a hypermucoviscosity phenotype (*rmpA/rmpA2*) [10]. Therefore, genomic studies could provide insights into the dynamics of dissemination of this bacterial complex, as well as the genes associated with antibiotic resistance and virulence.

Brazil is a continental country bordering most South American countries, and it is characterized by six major biomes, namely Amazon, Cerrado, Caatinga, Pantanal, Atlantic Forest, and Pampa, which is supposed to have the greatest biodiversity in the world [11]. In this country, KpSC has already been incriminated in nosocomial infections, including high-risk clones presenting multidrug resistance mainly due to the presence of carbapenemase genes, such as *bla*_NDM_ and *bla*_KPC_ [4,12,13,14,15,16,17]. Moreover, virulent and multidrug-resistant strains from natural environments and animals have also been reported [18,19,20]. Therefore, based on this scenario, we performed a genomic surveillance analysis on hundreds of KpSC genomes from Brazil, considering the three axes of the One Health concept (environment—animal—human), to reveal their resistome, virulome and genetic relationships in the context of this diverse scenario.

## 2. Materials and Methods

### 2.1. Public Data Set

Complete and draft genomes of *Klebsiella* isolated from Brazil (n = 378) were obtained from the National Center for Biotechnology Information (NCBI) on 29 October 2021: (https://www.ncbi.nlm.nih.gov/genome/browse/#!/prokaryotes/klebsiella). The accession numbers are supplied in Appendix A. 

### 2.2. Genome Sequencing and Assembly

In this study, we generated 21 *Klebsiella* spp. genomes, in which 14/21 were from the Brazilian Amazon (RR—Roraima), including animal and human/clinical isolates. The remaining genomes (n = 7) were human clinical ones from Rio de Janeiro (RJ). In this way, we were able to contribute genomic information from underrepresented Brazilian regions (Appendix A).

The genomic DNA extraction was done using a NucleoSpin Microbial DNA kit (Macherey-Nagel), and the genome libraries were constructed using Nextera paired-end libraries. The sequencing was performed by Illumina Hiseq 2500, generating reads of 250 bp length. The raw reads were filtered and trimmed using a NGS QC Toolkit v.2.3.3 [21], considering a Phred quality score ≥ 20. The genomes were de novo assembled with a SPAdes assembler v3.14.1 [22] or a A5-miseq pipeline v20160825 (IDBA-UD assembler) [23] and improved using a Pilon v1.23 [24]. 

### 2.3. Characterization of Klebsiella Genomes

The Kleborate v2.1.0 [25] pipeline was used for the genetic characterization of the *Klebsiella* genomes considering species designation, sequence typing, and identification of acquired virulence and antibiotic resistance genes. In addition, the Kleborate scored the virulence and antibiotic resistance profiles of the genomes based on the presence of clinically relevant gene markers. Virulence scores range from 0 to 5, based on the presence of three loci (yersiniabactin, colibactin, and aerobactin): yersiniabactin only (score 1), colibactin without aerobactin (score 2), aerobactin only (score 3), aerobactin and yersiniabactin without colibactin (score 4), and all three present (score 5). The resistance scores range from 0 to 3, based on the presence of genes associated with ESBL (score 1), carbapenemases (score 2), and carbapenemase plus colistin resistance (score 3). 

In addition, ABRicate v1.0.1 (https://github.com/tseemann/abricate accessed on 1 January 2022) was used for plasmid replicon typing (PlasmidFinder database) [26], and the identification of virulence (VFDB database) [27] and antibiotic resistance genes (CARD database) [28] in the putative plasmids. 

### 2.4. Phylogeny

The genomes were annotated by Prokka v1.12 [29] and their core genomes were estimated using Roary v3.13 [30]. Next, single nucleotide polymorphisms (SNPs) were extracted from the concatenated core genes by snp-sites v2.5.1 [31] and submitted to phylogenetic analysis using an IQTree v1.6.12 [32] to obtain a maximum likelihood tree with 1000 ultrafast bootstrap replicates [33]. The tree was generated using the iTOL web platform (https://itol.embl.de) accessed on 1 January 2022) [34].

## 3. Results

### 3.1. K. pneumoniae Species Complex Epidemiology and Sequence Typing

To establish the epidemiological links of the KpSC concerning the One Health axes, we performed a genomic analysis with new genomes as well as genomes available in the NCBI database. In total, 399 *Klebsiella* genomes from Brazil were analyzed, of which 21 were obtained in the present study and 378 from the NCBI (Appendix A). These genomes belonged to *Klebsiella* isolated from the environment, animals, and humans in every Brazilian region (CO, midwest; N, north; NE, northeast; S, south; and SE, southeast) (Figure 1) between 2003 and 2020 (Figure 2). 

Based on Kleborate taxonomic criteria, five *Klebsiella* species were identified, of which two had two subspecies: *Klebsiella pneumoniae* (n = 350), *Klebsiella quasipneumoniae* subsp. *quasipneumoniae* (n = 2), *Klebsiella quasipneumoniae* subsp. *similipneumoniae* (n = 16), *Klebsiella variicola* subsp. *variicola* (n = 20), *Klebsiella variicola* subsp. *tropica* (n = 1), *Klebsiella aerogenes* (n = 9), and *Klebsiella grimontii* (n = 1) (Appendix A). Most of the isolates (n = 349) were recovered from humans, while the others were from animals (n = 30; including vertebrates and invertebrates), and natural environments (n = 18; water, sewage, soil, plants) (Appendix A). All *K. aerogenes* (n = 9) genomes were from human/clinics and, therefore, were not included in the further analyses.

Most of the remaining 390 genomes belonged to *K. pneumoniae* (n = 350), encompassing many STs, some persisting for several years; most of them have been reported globally and, therefore, are pandemics, such as ST11 (n = 98), ST437 (n = 52), ST258 (n = 37), and ST340 (n = 27) (Figure 3, Appendix A). These four prevalent STs are part of the same clonal complex (CC258) and represent 61% (214/350) of the *K. pneumoniae* genomes from Brazil. Considering the Brazilian regions, ST11 was the most prevalent, except in the South (S) region, in which ST437 prevailed. However, as most of the regions are underrepresented (S, 15; NE, 27; N, 35; CO, 19) compared with the SE region (n = 288), this ST prevalence could be biased (Appendix A). Importantly, some STs, including pandemic ones, occurred in different sources (and sometimes in different regions): ST340, ST15, and ST11 in humans, animals (e.g., crabs and dogs), and environments (e.g., mangroves and water); ST307 in humans and animals (e.g., mussels, dogs, horses); ST437 in humans and environments (e.g., water); ST198 in humans and the environment (e.g., lettuce) (Appendix A). About the other species, *K. variicola* subsp. *variicola* was represented by multiple STs in humans (n = 9), animals (n = 8), and environments (n = 3), occurring in all Brazilian regions. In particular, ST355 was observed both in animals (e.g., *Bos indicus*) and in the environment (e.g., plants). In the same way, *K. quasipneumoniae* subsp. *similipneumoniae* was represented by several STs in the country’s regions and, interestingly, the ST1308 was observed in both humans and mosquitoes (Figure 3 and Appendix A). 

### 3.2. K. pneumoniae Species Complex Antibiotic Resistance and Virulence Determinants

The antibiotic resistance and virulence determinants of the KpSC genomes were surveyed using the Kleborate tool, which determines score values based on loci that contribute to clinically relevant antibiotic resistance (score from 1 to 3) or hypervirulence (score from 1 to 5) phenotypes. 

Regarding antibiotic resistance for *K. pneumoniae*, most genomes were classified in score 2, followed by 3, 1, and 0; for *K. variicola* subsp. *variicola*, most genomes were classified in score 0, followed by 2; for *K. quasipneumoniae* subsp. *similipneumoniae*, the genomes were classified in scores 0 or 2 (Figure 3 and Figure 4A, and Appendix A). 

In total, 336 genomes presented positive scores for β-lactamases, where few of them only harbor ESBLs (score 1; n = 59), while the others presented carbapenemases (score 2; n = 165), and carbapenemases with colistin resistance determinants (score 3; n = 112). Score 3 was only observed in *K. pneumoniae* species, including clinical and environmental species. Although genomes harboring only ESBLs (score 1) were outnumbered (n = 59), ESBLs also co-occurred in genomes carrying carbapenemases (53%; n = 213), where 128/165 genomes with score 2 had carbapenemase and ESBL (77%), and 85/112 genomes with score 3 had carbapenemase with colistin resistance determinants and ESBL (75%). Thus, most *Klebsiella* genomes co-harbor carbapenemase and ESBL genes.

The median number of antibiotic resistance genes contained in the genomes varied depending on the source, being 0, 5, and 10 genes in the animal, environmental and human genomes, respectively. In general, genomes of the *Klebsiella* species of the three sources presented heterogeneity of antibiotic resistance genes (Table 1) and resistance profiles (Table 2).

Some genomes from the same ST but occurring in different sources shared several antibiotic resistance genes, such as *cat*A1, *cat*B3, *sul*1, *bla*_CTX-M-2_, *bla*_CTX-M-15_, *bla*_OXA-1_, *bla*_OXA-2_, *bla*_KPC-2_, *aac*(3), *tet*(A), *tet*(D) (Appendix A). For instance: in ST11, GCA_013303005.1 genome (dog/2019) and GCA_011037475.1 genome (human/2016) shared *qnr*S1, *sul*1, *dfr*A12, *bla*_LAP-2_, *bla*_OXA-1_, *bla*_CTX-M-15_, *bla*_KPC-2_; in ST307, GCA_003194695.1 genome (horse/2017) and JALJAB000000000 genome (human/2017) shared *qnr*B1, *sul*2, *tet*(A), *dfr*A14, *bla*_TEM-1_, *bla*_OXA-1_, *bla*_CTX-M-15_. Most genomes analyzed (n = 342; 87%) were classified as MDR, since they harbored antibiotic resistance genes conferring resistance to ≥3 drug classes (Appendix A). Considering the three bacterial sources, differences in the proportion of MDR genomes were observed: human, 92% (314/340); environment, 66% (12/18); and animal, 46% (14/30). Concerning enzymatic resistance, among the carbapenemases, *bla*_KPC-2_ was the most prevalent, and the main alleles of ESBLs were *bla*_CTX-M-14_ and *bla*_CTX-M-15_ (Appendix A). The *sul* (sulfonamide resistance) and *dfr*A (trimethoprim resistance) genes were observed in 73% and 70% of the genomes, respectively, while to a lesser extent, *qnr* (quinolone resistance) and *tet* (tetracycline resistance) genes were found in 30% and 45% of the genomes, respectively (Appendix A). The *mcr* gene, involved in colistin resistance, was only observed in four clinical genomes: three of *K. pneumoniae* from different STs, and one from *K. quasipneumoniae* subsp. *similipneumoniae*. However, 110 genomes presented mutations in the *mgr*B and/or *pmr*B genes (genomes with score 3), also associated with colistin resistance (Appendix A). In addition, this in silico resistome investigation revealed the presence of the rare *bla*SCO-1 in five clinical genomes of *K. pneumoniae* ST11, ST307, ST392 (n = 2), and ST442, recovered from different regions (Appendix A). Moreover, based on class C beta-lactamase gene family sequences from the National Database of Antibiotic Resistant Organisms (NDARO), we searched *amp*C genes in all *Klebsiella* genomes, and none of them were detected.

The virulence loci analysis revealed that 146 genomes had score 1 (n = 75), score 2 (n = 68), score 3 (n = 2), and score 5 (n = 1), while 244 genomes presented the virulence score 0 (Figure 4B and Appendix A). Regarding *Klebsiella* species, there was only a high prevalence of genes associated with yersiniabactin and colibactin virulence loci in *K. pneumoniae* (Table 3), particularly in genomes belonging to the high-risk clone ST11 (Table 4). Interestingly, T6SS occurred in all species, as well as in all *K. pneumoniae* STs (Table 3 and Table 4). 

The virulence loci yersiniabactin and colibactin identified in animal, environmental, and human *K. pneumoniae* genomes were associated with different ICEKp (ICEKp3, 4, 10, 12). For instance, the ICEKp4, which carries the yersiniabactin 10 (*ybt* 10) loci, occurred in genomes from dogs and water (ST15), and mangroves and humans (ST11); and the ICEKp10, which carries the yersiniabactin 0 or 17 (*ybt* 0 and *ybt* 17) loci and the colibactin loci 3 (*clb* 3), occurred within ST11 genomes from crabs, water and humans (Appendix A). Most of *K. pneumoniae* belonging to ST11 had score 2, therefore harboring colibactin loci or colibactin plus yersiniabactin loci, the latter being the prevalent composition (Figure 3 and Appendix A). However, a deep analysis revealed that some of them had their loci annotated as truncated and/or incomplete (yersiniabactin, 17/144; colibactin, 20/69). The same occurred with other virulence loci (aerobactin, 2/3; salmochelin, 3/4; *rmp*ADC, 1/3) (Appendix A), hence were probably nonfunctional. Interestingly, some animal and environmental genomes were identified with positive virulence scores, such as 2 or 3 (Appendix A). 

### 3.3. K. pneumoniae Species Complex Plasmids

Using the ABRicate tool with the Plasmidfinder database, we predicted thousands of putative plasmid sequences with dozens of plasmid replicon types (n = 52) in 384 *Klebsiella* genomes of all species. It was not always possible to recover the entire plasmid sequence, as many genomes were highly fragmented, but it was possible to infer that some plasmid sequences were shared (considering coverage ≥ 70% and identity ≥ 80%) by strains from different STs, regions, and sources (Table 5). 

In addition, similar plasmids (also considering coverage ≥ 70% and identity ≥ 80%) could be observed being shared among different species, such as a group of small plasmids (~7.3–9.2 kb) present in *K. pneumoniae* (WERP01000003; water source), *K. quasipneumoniae* (VDFZ01000103; human source), and *K. variicola* (SZND01000045; human source). Additionally, a putative plasmid from *K. quasipneumoniae* (JAGTYC010000025; ~76 kb), isolated from a mosquito, presented an identity (82%) with a putative plasmid from *K. pneumoniae* (NTHU02000076; ~104 kb), isolated from a human. Curiously, the latter plasmid carried the *aph*(3), *aph*(6), and *sul*2 genes, while the *K. quasipneumoniae* plasmid did not. All putative plasmid sequences were screened for virulence and antibiotic resistance genes; however, due to the genome fragmentation, only sequences presenting these genes with any plasmid replicon gene in the same contig were considered. Among these 384 genomes, only five (four *K. pneumoniae* and one *K. grimontii*) presented putative plasmids with virulence genes (Appendix A). Interestingly, one of these plasmids (LYZC01000019.1; 148 kb) harbored seven core genes of the Type VI secretion system (*tss*B/*tss*C/*tss*K/*tss*L/*tss*D/*tss*H/*tss*I). In addition, it was possible to observe that two *K. pneumoniae* genomes (SPSP01000000.1 and SPSO01000000.1), both isolated from animals (*Callithrix penicillata*), harbored the pLVPK virulence plasmid (~220 kb; ~100% coverage), in fragmented contigs, encoding the aerobactin operon (*iuc*ABCD-*iut*A) together with the tellurium (*ter*) resistance operon, and *rmp*A2, a regulator gene of the mucoid phenotype of HvKP. Thus, this plasmid was responsible for the virulence score 3 of these genomes, since they do not encode either yersiniabactin or colibactin. Regarding antibiotic resistance genes, 224 plasmid sequences carried from one to 13 genes (Appendix A), where *bla*_KPC_ was the most common (n = 93). Interestingly, despite most of the KPC genes belonging to the same allele, several of them were in different putative plasmids and genomes of *K. pneumoniae* from different Brazilian regions. Other beta-lactamases were also found, such as *bla*_CTX-M_ (−2, −8, −9, and −15), *bla*_GES_ (−14), and *bla*_OXA_ (−1, −9, and −543), in addition to genes associated with resistance to aminoglycoside (*aac*, *aph*), fluoroquinolone (*qnr*), macrolide (*erm*B), chloramphenicol (*cat*), rifamycin (*arr*), sulfonamide (*sul*), colistin (*mcr*) (Appendix A). Some putative plasmids carrying antibiotic resistance genes were shared by different genomes from different sources and STs (Table 5). For instance, the putative plasmids LYMZ01000021 (animal, ST340) and JABBZB010000030 (human, ST11) harbored sequences encoding *bla*_LAP-2_ and *qnr*S1; JABSUB010000003 (animal, ST11) and JAEVGJ010000028 (human, ST437) harbored sequences encoding *bla*_KPC-2_ (Table 5). Thus, similar plasmids, carrying the same antibiotic resistance genes, are dispersed in strains from human, environmental, and animal sources. Importantly, there is a set of plasmids harbored by environmental genomes carrying enzymatic genes related to resistance to several antibiotics (Table 6).

## 4. Discussion

In 2019, the WHO established a group of critical priority pathogens associated with the worldwide spread of antimicrobial resistance [35]. This group of bacteria includes the remarkable pathogen *K. pneumoniae*, which together with other *Klebsiella* species constitutes the *K. pneumoniae* species complex (KpSC), which is strongly associated with the spread of antimicrobial resistance due to its high susceptibility to horizontal gene transfer [36], thus favoring the emergence of high-risk clones that can evolve into dominant clones and cause outbreaks. Furthermore, *Klebsiella* species are ubiquitous and can be found in a wide range of sources, such as humans, animals, natural environments, and foods [37]. In fact, we observed in the present analysis the same STs in different sources (e.g., *K. pneumoniae*: ST340, ST15, and ST11 in humans, animals, and the environment; *K. variicola*: ST355 in animals and the environment; *K. quasipneumoniae*: ST1308 in humans and animals), characterizing the zoonotic profile of these species. Furthermore, genomes from different sources were dispersed in the phylogeny, not forming groups by source. Therefore, there are no specific lineage associations with the source. Most environmental genomes were obtained from aquatic environments, which can be directly affected by the industrial, hospital, and sewage effluents, being potential “hot spots” for resistant bacteria and the exchange of genetic material between species [38]. In fact, we were able to identify some putative plasmids that were shared between different *Klebsiella* species, albeit from different sources. This suggests that the environment is not a barrier for *Klebsiella* species to exchange genetic material with each other.

Heterogeneity of *K. pneumoniae* STs has been reported in distinct Brazilian regions, mainly STs from the high-risk pandemic CC258 [16,39,40,41,42]. Based on the available *Klebsiella* spp. genomes, we observed that CC258 is prevalent in Brazil, considering the national level (61% of the *K. pneumoniae* genomes), mainly through the presence of ST11 (28% of the *K. pneumoniae* genomes). So far, ST11 has been mostly associated with outbreaks in Asia, mainly in China [43], while ST258 and ST512 are epidemics in Latin America and Europe [44]. However, in the present study, it was noticed that ST258 and ST512, together, represented only ~10% of the *K. pneumoniae* genomes (observed in three of the five Brazilian regions). Therefore, it is possible that ST11 outbreaks in Brazil are being underreported. Interestingly, this scenario can also be seen considering the animal and environmental *K. pneumoniae* genomes belonging to STs from CC258. 

An in-silico analysis, such as the one performed here, can also reveal information about genes that are not usually tracked, as occurred with *bla*SCO-1, which is supposedly rare in *K. pneumoniae* [45]; here, it was identified in some clinical genomes of *K. pneumoniae* of ST11, ST307, ST392, and ST442 that occur worldwide. Furthermore, several antibiotic resistance genes, identified in similar putative plasmids, were shared between genomes from different sources, suggesting, in general, a connection between these three One Health axes. Interestingly, a clear prevalence of yersiniabactin, colibactin, and T6SS was observed in ST11 genomes, including environmental and animal genomes, which may favor their success in causing outbreaks and invasive infections [46]. In addition to ST11, ST16 was another ST with a prevalence of the yersiniabactin loci, where about half of the analyzed ST16 genomes presented this locus. These genomes belong to a study that used a larval model to show the association of ST16 with high mortality rates when compared to ST11 [47].

In terms of South America, some comparisons can be made between *K. pneumoniae* data from Brazil and Colombia, a *bla*_KPC_-endemic country in South America [48]. In both countries, strains of CC258 prevail (mainly ST258/512 in Colombia and ST11 in Brazil) with high levels of MDR (~93%), but with differences in the rate of carbapenemase-producing strains, 91.5% in Colombia [48], and 75% in Brazil. Furthermore, in Colombia, there is no single prevalence of carbapenemases (*bla*_KPC-3_, 39%; *bla*_KPC-2_, 30%; *bla*_NDM-1_, 14%) [48], while *bla*_KPC-2_ is practically the only circulating allele (92%) in Brazil. Virulence was low among CC258 strains in both countries, with a similar prevalence of the *clb* 3 lineage (25.2% in Colombia and 28% in Brazil). On the other hand, the prevalence of yersiniabactin differed, with 76.9% of the CC258 isolates carrying this locus in Colombia (mainly *ybt* 17 and *ybt* 10) [48], and 44% in Brazil (mainly *ybt* 0 and *ybt* 9). 

Our analysis revealed a *K. quasipneumoniae* genome from the human/SE region belonging to ST334. This ST has been reported as a potential emerging outbreak-associated MDR clone [44], having already been identified in Pakistan, Cambodia, and Singapore [49]. Indeed, this Brazilian genome displayed a broad resistome, being in vitro confirmed as an XDR (data not shown). Two ST334 genomes from Singapore present a similar MDR profile but they carry distinct carbapenemases (*bla*_NDM-5_ and *bla*_OXA-181_) [49] about the Brazilian genome that carries *bla*_KPC-2_. These findings reveal the diversity of carbapenemase genes in this *Klebsiella* species.

Although *K. variicola* is commonly associated with plant ecosystems, infections have been reported in humans and less frequently in animals [50]. Here, we observed a wide range of sources, including soil, different animals, and humans, with most genomes presenting a score of 0, both for virulence and antibiotic resistance. Few genomes harbored carbapenemases (*bla*_GES-5_ and *bla*_KPC-2_), and ESBLs (*bla*_CTX_) (some in plasmids), showing that this *Klebsiella* species has the potential to develop the MDR profile [17].

Recently, several reports from Asia, Europe, and North America described the emergence of *Klebsiella* hypervirulent clones, in which there was a convergence of virulence and antibiotic resistance traits [9,10,51]. However, among Brazilian genomes, we could not observe this convergence (i.e., genomes presenting a virulence score ≥ 3 and an antibiotic resistance score ≥ 1; [25]). The exception was only one clinical genome (GCA_003326465.1) belonging to a new ST, but close to ST23 (it had a variant locus), and presenting a virulence score of 5 and antibiotic resistance score of 3. Indeed, ST23 is commonly associated with hypervirulent strains, as are other STs, such as ST37 [25,52]. The ST37 was observed here; however, it did not show any virulence trait, although it was predicted as MDR. The lack of convergence of virulence and antibiotic resistance was also observed in other *Klebsiella* species, as well as in genomes from animal and environmental sources.

As we have seen here that most of the population of KpSC in Brazil would be MDR, there could be a tendency for the appearance of hypervirulent clones since MDR clones circulating in clinical settings would be more likely to acquire virulence genes [10]. Indeed, the virulent plasmid pLVPK was identified here in two genomes of *K. pneumoniae* isolated from *Callithrix penicillata*, a commensal animal in urban areas [53]. Interestingly, these strains showed a virulent phenotype, but with an antibiotic susceptibility profile [54]. However, as *K. pneumoniae* is highly susceptible to horizontal gene transfer, these strains could show a convergent phenotype of virulence and resistance.

This study therefore contributes to the understanding of some underlying factors, resistome and virulome, driving the success of the *Klebsiella pneumoniae* species complex as a pathogen.

## Figures and Tables

**Figure 1 microorganisms-10-02281-f001:**
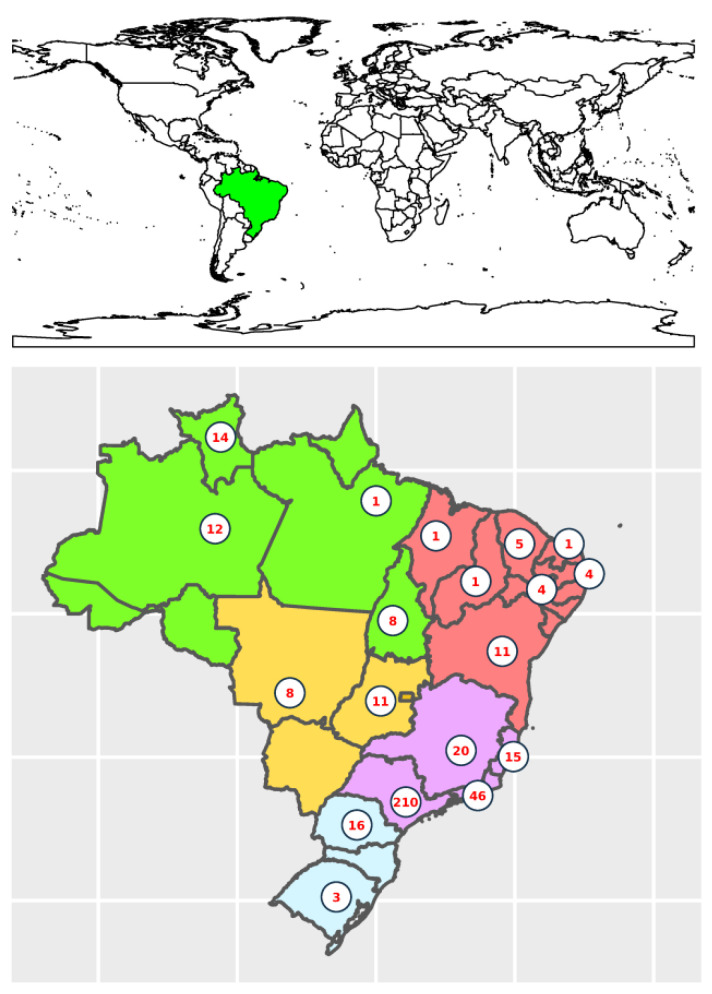
Brazilian regions and distribution of *Klebsiella* genomes. The values represent the amount of *Klebsiella* genomes obtained from each Brazilian state. The Brazilian regions are colored in green (North), red (Northeast), yellow (Midwest), lilac (Southeast), and blue (South).

**Figure 2 microorganisms-10-02281-f002:**
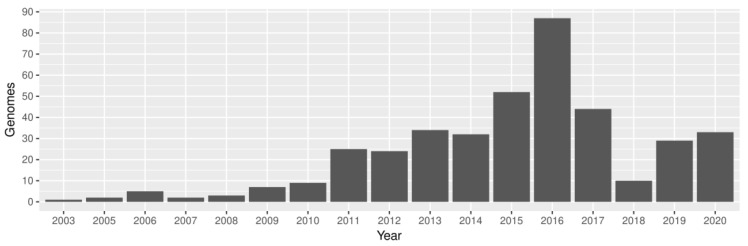
Number of sequenced and available *Klebsiella* genomes (*y* axis) by year of isolation (*x* axis).

**Figure 3 microorganisms-10-02281-f003:**
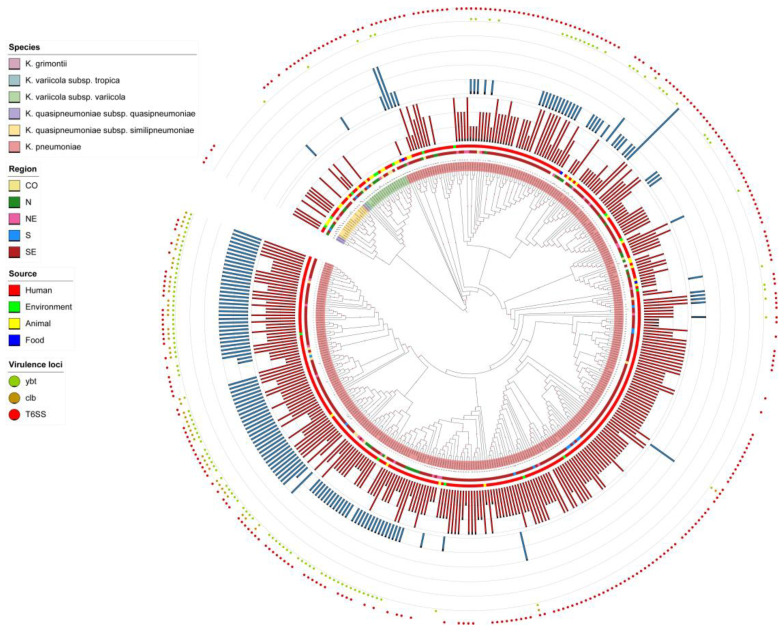
Maximum-likelihood tree based on the core genome of *Klebsiella* spp. The different species are indicated by the background colors of the accession numbers. The ST number of each genome is next to the accession number. There are two orbits of colored blocks, where the innermost represents the Brazilian regions from where the genome was obtained, and the outermost represents the source of isolation. More external to these blocks, the different sizes of the thin bars (0 to 5) indicate the antibiotic resistance (red) and virulence (blue) scores of the genomes. Outer-colored circles indicate genomes with complete operons of yersiniabactin (green), and colibactin (beige) loci; and the presence of Type VI secretion system gene clusters (red).

**Figure 4 microorganisms-10-02281-f004:**
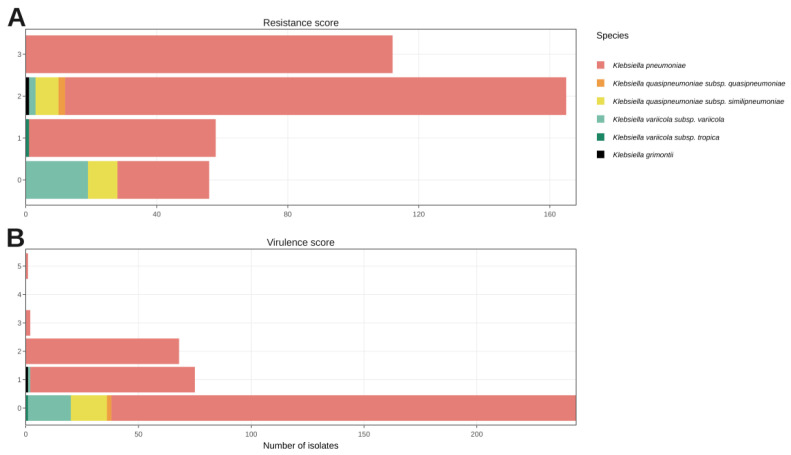
Antibiotic resistance (**A**) and virulence (**B**) scores of KpSC.

**Table 1 microorganisms-10-02281-t001:** Antibiotic resistance genes identified in the genomes of *Klebsiella* species.

Species	AGly	Col	Flq	MLS	Phe	Rif	Sul	Tet	Tmt	Bla	Bla_ESBL	Bla_Carb
*K. quasipneumoniae*	*aac aad*A *aph rmt*D *str*A *str*B		*qnr*A *qnr*B *qnr*E	*ere*A	*cat*B		*sul*	*tet*D	*dfr*A	*oxa tem*	*ctx*	*ges kpc ndm*
*K. variicola*	*aac aad*A *sat*2 *str*A *str*B		*qnr*B		*cat*A		*sul*		*dfr*A	*oxa tem*	*ctx*	*ges kpc*
*K. pneumoniae*	*aac aad*A *ant aph arm*A *rmt*B *rmt*B *rmt*C *rmt*G *sat*2 *str*A *str*B	*mcr*	*qnr*A *qnr*B *qnr*E *qnr*S *qnr*VC	*ere*A *erm*B *mph*A *mph*E	*cat*A *cat*B *cat*II *cml*A *flo*R	*arr*	*sul*	*tet*A *tet*B *tet*D *tet*G	*dfr*A	*lap oxa shv* *sco tem*	*ctx*	*imp kpc ndm*

AGly, aminoglycosides; Col, colistin; Flq, fluoroquinolone; MLS, macrolides; Phe, phenicols; Rif, rifampin; Sul, sulfonamides; Tet, tetracyclines; Tmt, trimethoprim; Bla, β-lactamases; Bla_ESBL, ESBL extended-spectrum β-lactamases; Bla_Carb, carbapenemases.

**Table 2 microorganisms-10-02281-t002:** Number of *Klebsiella* genomes carrying antibiotic resistance genes to different classes of antibiotics per species.

Species	#	AGly	Col	Flq	MLS	Phe	Rif	Sul	Tet	Tmt	Bla	Bla_ESBL	Bla_Carb	# MDR
*K. quasipneumoniae*	18	11 (61%)	1 (5%)	6 (33%)	1 (5%)	7 (38%)	0	9 (50%)	1 (5%)	8 (44%)	7 (38%)	7 (38%)	9 (50%)	10 (55%)
*K. variicola*	21	3 (14%)	0	1 (4%)	0	1 (4%)	0	2 (9%)	0	2 (9%)	2 (9%)	2 (9%)	2 (9%)	3 (14%)
*K. pneumoniae*	350	113 (32%)	3 (<1%)	113 (32%)	194 (55%)	225 (64%)	20 (5%)	277 (79%)	177 (50%)	267 (76%)	278 (79%)	262 (74%)	265 (75%)	328 (93%)

AGly, aminoglycosides; Col, colistin; Flq, fluoroquinolone; MLS, macrolides; Phe, phenicols; Rif, rifampin; Sul, sulfonamides; Tet, tetracyclines; Tmt, trimethoprim; Bla, β-lactamases; Bla_ESBL, ESBL extended-spectrum β-lactamases; Bla_Carb, carbapenemases; #, number.

**Table 3 microorganisms-10-02281-t003:** Prevalence of virulence loci in *Klebsiella* species in Brazil.

	#	*ybt*	*clb*	*iuc*	*iro*	*rmp*ADC	T6SS
*pneumoniae*	350	142 (40%)	69 (19%)	3 (<1%)	4 (<1%)	3 (<1%)	268 (76%)
*K. quasipneumoniae*	18	0	0	0	0	0	4 (22%)
*K. variicola*	21	1 (<1%)	0	0	0	0	18 (85%)

Ybt, yersiniabactin; Clb, colibactin; Iuc, aerobactin; Iro, salmochelin; rmpADC, hypermucoidy loci; #, number.

**Table 4 microorganisms-10-02281-t004:** Prevalence of virulence loci in the main STs of *Klebsiella pneumoniae* in Brazil.

ST	#	*ybt*	*clb*	*iuc*	*iro*	*rmp*ADC	T6SS
11	98	90 (91%)	59 (60%)	0	0	0	67 (68%)
437	52	2 (<1%)	2 (<1%)	0	0	0	48 (92%)
258	37	0	0	0	0	0	21 (56%)
340	27	3 (11%)	1	0	0	0	12 (44%)
16	25	12 (48%)	0	0	0	0	25 (100%)
15	18	3 (16%)	0	0	0	0	18 (100%)

Ybt, yersiniabactin; Clb, colibactin; Iuc, aerobactin; Iro, salmochelin; rmpADC, hypermucoidy loci; #, number.

**Table 5 microorganisms-10-02281-t005:** Putative plasmids shared between *Klebsiella* genomes.

Accession Number	Size (bp)	Source	Region	Year	ST
NCVW01000066	140.347	Animal	SE/SP	2016	307
PPHO01000091	140.347	Arugula	SE/SP	2016	2739
LZCY01000051	94.582	River	SE/SP	2011	437
QXMY01000007	119.474	Human	SE/SP	2010	11
JABBZB010000030	64.338	Human	SE/SP	2018	11
LYMZ01000021	88.051	Animal	SE/MG	2012	340
JAAQON010000029	66.917	Meat	N/PA	2019	3827
JABBZB010000035	48.305	Human	SE/SP	2018	11
JABUOR010000039	47.331	Lake	SE/SP	2017	15
JABSUB010000003	54.609	Animal	SE/SP	2019	11
JAEVGJ010000028	53.207	Human	SE/SP	2014	437
JABBZC010000039	12.764	Human	SE/SP	2014	340
JABEPG010000069	12.143	Animal	SE/SP	2017	11
MJBQ01000030	12.096	Lake	SE/SP	2011	340
JAEVGO010000022	54.438	Human	SE/SP	2014	437
KX062091	52.536	Urban river	SE/SP	2011	437
QXMZ01000002	52.364	Human	SE/SP	2009	437

**Table 6 microorganisms-10-02281-t006:** Antibiotic resistance genes identified in putative plasmids from *Klebsiella* environmental genomes.

Putative Plasmids	Source	Region	Species	ST	*aac*(6′)-Ib8	*aac*(6′)-Ib9	*ant*(3″)-IIa	*ges*-5	*kpc*-2	*oxa*-9	*shv*-134	*shv*-5	*tem*-181	*dfr*22	*dfr*A30	*qac*E	*rmt*D	*sul*1
CP066860.1	Sewer effluent	S	*K. quasipneumoniae*	ST5527		X	X	X		X		X	X	X			X	X
CP076869.1	Urban lake	CO	*K. pneumoniae*	ST5236					X									
WERP01000062.1	Water	SE	*K. pneumoniae*	ST4416														X
CP067435.1	Sewer effluent	S	*K. grimontii*	NA				X										X
CP067436.1								X										
CP067439.1					X													
CP067440.1											X							
LZCZ01000029.1	Urban river	SE	*K. pneumoniae*	ST437											X	X		
NSLG01000092.1	Urban lake	SE	*K. pneumoniae*	ST11											X	X		
WERN01000017.1	Water	SE	*K. pneumoniae*	ST661														X
WERO01000020.1	Water	SE	*K. pneumoniae*	ST4415														X

(X): gene presence.

## Data Availability

Genome information is detailed in Appendix A.

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
