# Peer review of "Genomics of *Klebsiella pneumoniae* Species Complex Reveals the Circulation of High-Risk Multidrug-Resistant Pandemic Clones in Human, Animal, and Environmental Sources"

_microorganisms, 2022, doi:10.3390/microorganisms10112281_

Round 1
Reviewer 1 Report
The present study analyzed the genomics of Klebsiella pneumoniae species complex isolated from Brazil. The authors analyzed 390 genomes of KpSC, 378 from public DB and 21 from this study, obtained from human, animal, and environment. Of the 5 Klebsiella species analyzed, the most prevalent species is K. pneumoniae (n=350). The manuscript demonstrated that clonal distribution based on the Brazilian regions in association with resistance genes and virulence determinants. This study can provide insight into the understanding of evolutionary change of MDR and virulent KpSC. The authors well-analyzed epidemiological marker (STs), resistance genes, and virulence determinants of KpSC. This work is interesting, but needs clarification about the following points.
1. The legend of Figure 2 is confusing whether year in X-axis means isolation year of isolates or year of genome sequenced.
2. Figure 3 should be deleted because the distribution of species is described in the text (line 122-125).
3. Table 1 should be revised. All gene names are presented based on species. Gene names should be classified by antibiotic classes because Table 2 shows the prevalence of resistance genes by different classes of antibiotics.
4. The number of genes in Table 2 is different from Table S1. Count the number of genes carefully.
5. All bacterial name should be italic characters.
6. Line 36. MDR is first appeared and should be expressed in full name. Thereafter, multidrug resistance should be abbreviated.
7. Line 41. Nosocomial should be replaced to human.
Reviewer 2 Report
In the manuscript “Genomics of Klebsiella pneumoniae species complex reveals the circulation of high-risk multidrug-resistant pandemic clones in human, animal, and environmental sources”, the authors performed a genomic surveillance analysis of hundreds of Klebsiella pneumoniae species complex (using all public genomes from Brazil plus 21 generated in this study) to reveal their resistome, virolome and genetic relationships. The Klebsiella genomes were characterized using the Kleborate pipeline, however no association was established between the genomic profiles and the phenotypes of the strains. A global and interesting analysis of the genomes is carried out by the authors; however the hypotheses are too generic, reducing the impact and novelty of the results obtained. It is unclear why the 21 genomes were generated, why they were included in the analysis, and what their relevance to this study is. The predominance of genomes from the southeast region of Brazil, and of isolates recovered from humans end up biasing the results obtained. One possibility would be to analyze these data by grouping the samples and then comparing the groups.
Minor comments: Species and gene names should be italicized throughout the document, acronyms defined; and image quality and resolution improved (especially Figure 4).
Reviewer 3 Report
A very interesting manuscript analyzing the genome of Klebsiella pneumoniae species complex in Brazil from the point of view of the One Health concept is presented. The problem of antimicrobial resistance (AMR) undoubtedly represents one of the most serious problems of contemporary medicine and at the same time belongs to the priorities of the WHO. I therefore rate the submitted article as very current and beneficial.
However, I have some comments on the text, see below, and unfortunately I cannot recommend the article for acceptance in its current form. I believe that after incorporating the comments raised, the article will be suitable for acceptance and I will be very happy to recommend it (based on a new reading of the text).
Comments:
1) Names of bacteria are given in italics, in the text these names are given inconsistently, both in italics and in normal font.
2) It is correctly stated in the introduction that KpSC resistance to beta-lactam antibiotics (penicillins, cephalosporins and carbapenems) represents a significant problem in the treatment of infections caused by resistant strains. However, I do not understand why only the genes encoding KPC, MBL and OXA enzymes (as the most common) are listed and not ESBL and AmpC. I believe that especially the occurrence of ESBL is equally important in KpSC and in general even more frequent than the occurrence of the mentioned enzymes. I consider it necessary to reformulate this part of the introduction.
3) I do not understand why the resistance scores (defined as 1 to 3) do not include the carbapenemases plus ESBL variant. The results confirm that the frequency of ESBLs, for example in the case of Klebsiella pneumoniae, is almost identical to the frequency of carbapenemases. I recommend supplementing this score.
4) It is stated in the results that most genomes showed carbapenemases and carbapenemases with colistin resistance. In my opinion, this statement is not entirely accurate, resistance caused by ESBL production is more common than colistin resistance.
5) Why is the prevalence of colistin resistance genes not listed in Table 2? Especially in connection with the claim that carbapenemases with colistin resistance is very common.
6) Why were genes tested for resistance to rifamycins, which are not drugs of choice in infections with an etiological role of KpSC?
7) Why are genes encoding AmpC beta-lactamases not included in the analysis?
Round 2
Reviewer 2 Report
Dear authors,
Thank you for your reply. The changes made improved the manuscript. However, in my opinion, the hypotheses remain generic and the novelty of the results is limited. What was the inclusion criterion for the authors to have sequenced the 21 Klebsiella genomes described in this study? Was it just to increase the number of genomes analyzed or is there another reason?
Author Response
We have added a paragraph in the methodology about the newly sequenced genomes:
In this study, we generated 21 Klebsiella spp. genomes, in which 14/21 were from the Brazilian Amazon (RR - Roraima), including animal and human/clinical isolates. The remaining genomes (n=7) were human clinical from Rio de Janeiro (RJ). In this way, we contribute with genomic information from underrepresented Brazilian regions (Table S1).
Reviewer 3 Report
First of all, I thank the authors of the manuscript for editing the text based on my comments.
The text has been adequately edited and I am now pleased to recommend the manuscript for acceptance.
Author Response
We thank you for the review.